# The Advanced Diabetes Technologies for Reduction of the Frequency of Hypoglycemia and Minimizing the Occurrence of Severe Hypoglycemia in Children and Adolescents with Type 1 Diabetes

**DOI:** 10.3390/jcm12030781

**Published:** 2023-01-18

**Authors:** Tatsuhiko Urakami

**Affiliations:** Department of Pediatrics and Child Health, Nihon University School of Medicine, Tokyo 173-8610, Japan; urakami.tatsuhiko@nihon-u.ac.jp; Tel.: +81-3-3972-8111; Fax: +81-3-3958-5744

**Keywords:** hypoglycemia, severe hypoglycemia, children, adolescents, type 1 diabetes, advanced diabetes technology

## Abstract

Hypoglycemia is an often-observed acute complication in the management of children and adolescents with type 1 diabetes. It causes inappropriate glycemic outcomes and may impair the quality of life in the patients. Severe hypoglycemia with cognitive impairment, such as a convulsion and coma, is a lethal condition and is associated with later-onset cognitive impairment and brain-structural abnormalities, especially in young children. Therefore, reducing the frequency of hypoglycemia and minimizing the occurrence of severe hypoglycemia are critical issues in the management of children and adolescents with type 1 diabetes. Advanced diabetes technologies, including continuous glucose monitoring and sensor-augmented insulin pumps with low-glucose suspension systems, can reduce the frequency of hypoglycemia and the occurrence of severe hypoglycemia without aggravating glycemic control. The hybrid closed-loop system, an automated insulin delivery system, must be the most promising means to achieve appropriate glycemic control with preventing severe hypoglycemia. The use of these advanced diabetes technologies could improve glycemic outcomes and the quality of life in children and adolescents with type 1 diabetes.

## 1. Introduction

Hypoglycemia is an often-observed acute complication in the management of children and adolescents with type 1 diabetes. It causes inappropriate glycemic outcomes and may impair the quality of life in the patients [1]. Reducing hypoglycemia is an important objective that can be attained by evaluating the risk factors for problematic hypoglycemia and by introducing advanced diabetes technologies to the management of diabetes [2].

Severe hypoglycemia is defined as a condition with serious cognitive dysfunction requiring external help from other persons [3]. Severe hypoglycemia is still a lethal condition and is demonstrated to be the cause of death in 4–10% of pediatric patients with type 1 diabetes [4,5,6]. It may cause permanent brain damage, cognitive impairment and brain-structural abnormalities, especially in young children with type 1 diabetes [7,8,9,10,11,12,13,14]. Table 1 shows changes in the incidence of severe hypoglycemia in children and adolescents with type 1 diabetes over time. A high incidence of severe hypoglycemia was demonstrated by the Diabetes Control and Complications Trial (DCCT) in 1997 [15], i.e., 61.2/100 persons/year in patients with intensive insulin treatment (multiple injections of insulin with basal-bolus regimen) and 18.7/100 persons/year in those with conventional insulin treatment (insulin injections twice a day with a mixture of regular- and intermediate-acting insulin), respectively. High incidence was also reported in the large pediatric cohorts in Australia [16] and Colorado, USA [17] in the early 2000s. However, the incidence markedly declined over time, eventually resulting in below 6.0/100 persons/year [18,19,20,21]. Development in the management of diabetes might contribute to reducing the occurrence of severe hypoglycemia. The advanced diabetes technologies in recent years could enable improved glycemic control by decreasing the risk of severe hypoglycemia [22,23,24]. Such advances include the introduction of new insulin analogs, increased frequency of self-monitoring of blood glucose (SMBG), use of continuous glucose monitoring (CGM) and insulin pump therapy. However, despite these new technologies, severe hypoglycemia is still a fear for pediatric patients with type 1 diabetes, family members and their caregivers [3,25].

## 2. Morbidity of Severe Hypoglycemia (Table 2)

### 2.1. Neurological Outcomes

Meta-analyses of the studies demonstrated that children with type 1 diabetes were likely to show impaired overall intellectual function as compared with healthy control subjects, and that the domains of executive functions, learning memory and processing speed were also impaired [26,27]. Several studies in children with type 1 diabetes showed that frequent episodes of severe hypoglycemia were associated with worse performance as compared with healthy control subjects on certain attentions, including overall cognitive function and verbal and visual memories. Younger age of onset, being under 6 years of age, and having frequent episodes of severe hypoglycemia especially contributed to causing cognitive impairment and delayed brain development [7,9,10,11]. Furthermore, severe hypoglycemia with a convulsion played a role in greater performance deficits, including overall cognitive function, attention tasks, and verbal and visual memories [28], while other studies indicated that cognitive function was more seriously impaired by hyperglycemia than by hypoglycemia [8,29,30,31]. It was reported that the degree of exposure to hyperglycemia was associated with full-scale IQ scores and executive functioning, and a long period of hyperglycemia may contribute to cognitive impairment in children with type 1 diabetes [31]. Therefore, hyperglycemia as well as hypoglycemia may be associated with cognitive impairment.

**Table 2 jcm-12-00781-t002:** Main consequences of severe hypoglycemia.

Neurological outcomes
- Impaired intellectual function including overall IQ, executive functions, learning memory and processing speed
- Worse performance including cognitive function, attention tasks, and verbal and visual memories
- Brain-structural abnormalities including greater hippocampal volumes and reduced gray- and white-matter volumes
Psychological outcomes
- Anxiery, increase general fatigue, insufficient sleep, and impairment of the quality of life

The association of brain-structural abnormalities accompanied by severe hypoglycemia was reported in children with type 1 diabetes. Severe hypoglycemia is likely to harm neurons in the medial temporal region, including the hippocampus [3]. Greater hippocampal volumes [13] and reduced gray and white matter volumes were seen in children who experienced convulsions with hypoglycemia [11]. Moreover, another study demonstrated that neurological changes in gray and white matter occurred not only with hypoglycemia, but also with hyperglycemia [14]. The relation between frequent episodes of severe hypoglycemia and the increased risk of later-onset epilepsy was also reported [32,33].

### 2.2. Psychological Outcomes

Fear of severe hypoglycemia may cause considerable anxiety, which can make a negative impact on daily activities and the management of diabetes [34]. In children with type 1 diabetes, family members and their caregivers may suffer from increased general fatigue, insufficient sleep, and impairment of their quality of life [35,36]. Fear of severe hypoglycemia, especially during sleeping time, is a most serious problem in young children with type 1 diabetes and family members [37]. While this fear can cause anxiety, avoidance of hypoglycemia may be adaptive, leading to appropriate vigilance in glucose management.

## 3. Risk Factors for Developing Severe Hypoglycemia

Possible risk factors for developing severe hypoglycemia in children and adolescents with type 1 diabetes are shown in Table 3.

### 3.1. Younger Age

Several studies have shown that young children are likely to experience hypoglycemia more frequently and/or more seriously as compared with adults with type 1 diabetes because they have more physical activities, unstable eating habits and irregular lifestyles, and are unable to communicate symptoms of hypoglycemia [38,39,40,41,42]. In addition, they tend to impair counterregulatory hormone responses to subsequent hypoglycemia via autonomic nerve function [41]. Neurological damage caused by severe hypoglycemia is more frequently and/or more seriously observed in young children with type 1 diabetes [12,13]. 

### 3.2. Nocturnal Hypoglycemia

Counterregulatory hormone responses to hypoglycemia attenuate in sleeping time [41,43], and patients with type 1 diabetes are likely to be less awakened by hypoglycemia as compared with healthy control subjects [41]. Several studies have shown the frequency of nocturnal hypoglycemia as 15–25% during the night in children with type 1 diabetes [44,45]. The Juvenile Diabetes Research Foundation (JDRF) continuous glucose monitoring (CGM) study group in 2010 reported frequent and prolonged nocturnal hypoglycemia on 8.5% of nights in both children and adults but more prolonged episodes in children. In this study, the mean time spent in nocturnal hypoglycemia was 81 min [45]. Almost half of these hypoglycemic events were unrecognized by patients themselves, family members and their caregivers [46,47]. Monitoring of overnight glucose levels is recommended, especially if patients have additional risk factors that may predispose them to nocturnal hypoglycemia.

### 3.3. Impaired Awareness of Hypoglycemia

In healthy people without diabetes, endogenous insulin secretion is closed off, and counterregulatory hormones (glucagon, epinephrine, and norepinephrine) are released in response to hypoglycemia. However, in patients with type 1 diabetes, there is a progressive loss of glucagon response to insulin-induced hypoglycemia. Impaired awareness of hypoglycemia, one of the acquired complications with insulin treatment, can be caused by defective counterregulatory hormone responses to hypoglycemia. It is observed as early as 12 months after the onset of diabetes [48,49]. This condition was observed in approximately a quarter of adults with type 1 diabetes, while in children and adolescents, a similarly high prevalence (33%) was reported in 2002, which decreased to 21% in 2015 [50,51]. Although the prevalence of impaired awareness of hypoglycemia has declined, it remains a concern in a substantial proportion of children and adolescents with type 1 diabetes. Patients with impaired awareness of hypoglycemia showed a six-fold increase in the frequency of severe hypoglycemia [52]; therefore, it should be necessary to evaluate this condition as a part of clinical management. Impaired awareness of hypoglycemia can be reversed by avoiding the occurrence of hypoglycemia for 2 to 3 weeks [53]; however, this may be difficult to accomplish with current insulin treatment. 

### 3.4. Frequent Episodes of Hypoglycemia

The majority of children with type 1 diabetes experience isolated episodes of severe hypoglycemia; however, some experience recurrent episodes of severe hypoglycemia. Frequent episodes of hypoglycemia contribute to defective counterregulatory hormone responses to a subsequent decline in glucose levels. Therefore, prior episodes of frequent hypoglycemia may play an important role in developing subsequent severe hypoglycemia [3]. After an episode of severe hypoglycemia, the risk of recurrent severe hypoglycemia remains higher for up to 4 years as compared with patients who have never experienced severe hypoglycemia [54]. Defective counterregulatory hormone responses and impaired awareness of hypoglycemia contribute to hypoglycemia-related autonomic nerve failure, resulting in subsequent severe hypoglycemia [55]. Rarely, self-administration of insulin causes repeated and unexplained severe hypoglycemia and should be considered a sign of psychological disorders (Factitious hypoglycemia) [56]. 

### 3.5. Glycemic Control

Previous studies reported that strict glycemic control was associated with an increased frequency of severe hypoglycemia, especially in young children with type 1 diabetes [15,16,21]. The DCCT demonstrated that there was a threefold increased risk of severe hypoglycemia in individuals requiring intensive management with lower HbA1c levels as compared with those treated conventionally [15]. However, recent studies have demonstrated that the association between glycemic control and the risk of severe hypoglycemia seems to be weakened accompanied by an improvement in diabetes management with a decreased risk of severe hypoglycemia [16,18,21,22,23,57,58]. In the Type 1 Diabetes Exchange (T1D Exchange) and the Diabetes-Patienten-Verlaufsdokumentation (DPV) registry, there were no increased rates of hypoglycemic coma in those aged less than 6 years with HbA1c levels of below 7.5% (58.5 mmol/moL) as compared with those with higher HbA1c levels [57]. No differences in HbA1c levels were also reported from an Indian study assessing children with or without severe hypoglycemia [58]. The decrease in the frequency of severe hypoglycemia may have resulted from advanced insulin treatment and glucose monitoring and improved hypoglycemia education during the past decades. Low HbA1c levels may be not a predictable indicator for severe hypoglycemia in children and adolescents with type 1 diabetes [22]. Therefore, optimal glycemic control by appropriate diabetes management can be achieved without an increase in the frequency of severe hypoglycemia.

## 4. Advanced Diabetes Technologies for Reduction in the Frequency of Hypoglycemia 

There are a variety of technological devices available to reduce hypoglycemia. However, the choice of device must be a decision based on a dialogue between the care providers and the patients with diabetes [3]. Currently, better peakless basal insulin analogs with more constant absorption are available, which can decrease the fluctuation of glycemia and hypoglycemic events. On the other hand, advanced diabetes technologies including an insulin pump and CGM are more useful tools for reducing hypoglycemia. A meta-analysis demonstrated that the use of an insulin pump in pediatric patients may be superior to insulin injections in reducing the incidence of severe hypoglycemia [59]. As compared with insulin injections, a lower risk of severe hypoglycemia was associated with an insulin pump in the T1D Exchange [60], the DPV registry [61] and the International Pediatric SWEET Registry [62]. It has been reported that the use of CGM [63] or the sensor-augmented insulin pump (SAP) with the control algorithms including the low glucose suspend (LGS) system [64,65] could prevent severe hypoglycemia in patients with impaired awareness of hypoglycemia. Prevention of severe hypoglycemia remains a most important critical issue in the management of children and adolescents with type 1 diabetes [3]. The closed-loop system seems to be the best means for the reduction of hypoglycemia; however, in the first step towards the closed-loop system, an insulin pump and CGM enabled children with type 1 diabetes to further reduction of hypoglycemia [3]. Possible diabetes technologies to prevent the occurrence of severe hypoglycemia and to reduce the risk of developing severe hypoglycemia are shown below.

### 4.1. CGM

CGM has recently taken the place of SMBG by providing real-time glucose levels and glucose trends. The data from CGM allow the review of glucose variability and the identification of asymptomatic hyper/hypoglycemia. The JDRF reported frequent and prolonged events of hypoglycemia, especially during nighttime, in children and adolescents with type 1 diabetes using CGM, i.e., hypoglycemia was found in 8.5% during the nighttime and the duration of hypoglycemia beyond 2 h was 23% during the nighttime [45]. We also found a higher frequency of hypoglycemia during the nighttime in children with type 1 diabetes using CGM, i.e., the mean frequency of time below range (TBR: glucose level of less than 70 mg/dL [3.9 mmol/L]) [66] was significantly greater in the 0–6 h (16.9% ± 5.2%) than in 6–12 h and 18–24 h time periods (7.8% ± 2.9%, 6.8% ± 4.8%; *p* < 0.01, respectively) [67]. On the other hand, several studies have demonstrated that CGM can reduce hypoglycemia concomitants with an improvement of HbA1c levels in patients with type 1 diabetes [60,68,69,70,71]. The randomized, controlled multicenter study showed a reduction of TBR concomitant with a decrease in HbA1c levels in children and adolescents as well as adults with type 1 diabetes [69]. There was a tendency to reduce severe hypoglycemia in the DPV registry and the T1D Exchange with CGM initiation [60,71], while another study showed that this effect was not elucidated in pediatric patients with type 1 diabetes [72]. 

There are currently two types of CGM, i.e., intermittently scanned CGM (isCGM) and real-time CGM (rtCGM). Unlike isCGM, rtCGM has high- and low-glucose alert/alarm systems, which warn patients and their caregivers of immediate or prolonged events of hyperglycemia or hypoglycemia. Various comparative studies have demonstrated that rtCGM is superior to isCGM for the reduction of hypoglycemia and improvement of glycemic control in adults with type 1 diabetes [73,74,75,76]. We also found that rtCGM had more beneficial effects for an increase of time in the range (TIR: glucose level of 70–180 mg/dL (3.9–10 mmol/L) [66] with a marked decrease of TBR compared with isCGM in children with type 1 diabetes (TBR: 4.3% ± 2.7% in rtCGM vs. 10.2% ± 5.4% in isCGM; *p* < 0.001) [77]. The real-time alert/alarm system may be the major reason for the greater efficacy of rtCGM in reducing hypoglycemia. Moreover, Ly et al. [78] reported that rtCGM with the low-glucose alarm system improved epinephrine response, and rtCGM might be effective for reducing impaired awareness of hypoglycemia and possibly avoiding the occurrence of severe hypoglycemia in children and adolescents with type 1 diabetes. 

### 4.2. Sensor-Augmented Insulin Pump with Low Glucose Suspension System and Predictive Low Glucose Suspension System

It has been reported that the use of an insulin pump can reduce nocturnal hypoglycemia [79], and this is further reduced by SAP with the control algorithms, which can suspend the basal insulin infusion with sensor-detected [64] or sensor-predicted hypoglycemia [80]. If the pump users do not recognize the warning sounds, the LGS system automatically suspends the basal insulin infusion for up to 2 h in response to sensor-detected hypoglycemia, after which the basal insulin infusion is resumed at the programmed rate. SAP with the LGS system can decrease moderate-to-severe hypoglycemia, especially during nighttime [65]. Predictive low-glucose suspend (PLGS) systems include MiniMed 640G (Medtronic, Northridge, CA, USA) and Tandem Basal IQ (Tandem Inc., San Diego, CA, USA). MiniMed 640G suspends the basal insulin infusion with the hypoglycemia prediction algorithm. Basal insulin infusion is usually suspended when the sensor glucose level is below 70 mg/dL (3.9 mmol/L) above the user-set low-glucose limit and is predicted to be 20 mg/dL (1.1 mmol/L) above this low limit for 30 min. When the users do not interfere, the insulin delivery resumes after the suspension of two hours or less at the programmed rate (Figure 1). The use of SAP with the PLGS system more effectively reduces the frequency of hypoglycemia and the risk of developing severe hypoglycemia in patients with type 1 diabetes [64,81,82,83,84,85,86,87]. Time in hypoglycemia (the glucose level of less than 63 mg/dL (3.5 mmol/L)) was reduced from 2.8% at baseline to 1.5% during the 6-month study of the PLGS system as compared with a reduction from 3.0% to 2.6% with SAP without the PLGS system, representing a close to 50% reduction in hypoglycemia [64]. Another study also found the superiority of the PLGS system in both adults and adolescents and children with type 1 diabetes, i.e., the median time of TBR (glucose level less than 70 mg/dL (3.9 mmol/L)) was reduced from 3.6% at baseline to 2.6% during the 3-week period in the PLGS system as compared with 3.2% in SAP without the PLGS system (difference (the PLGS system—SAP) = −0.8%, 95% CI −1.1 to −0.5; *p* < 0.001) [87]. 

### 4.3. Hybrid Closed-Loop System

An automated insulin delivery system allows for the automatic adjustment of basal insulin infusion based on the sensor glucose levels to prevent out-of-range high and low glucose concentrations; however, the pump users must administer bolus insulin doses according to the carbohydrate consumptions at each meal and for corrections of glucose levels (Figure 2). Therefore, the current automated insulin delivery system is referred to as the hybrid closed-loop system. Medtronic 670G/770G (Medtronic, Northridge, CA, USA) is approved for use in children above 7 years of age, Control IQ (Tandem Inc., San Diego, CA, USA) for children above 6 years of age, and CamAPS FX interoperable application (CamDiab, Cambridge, UK) for children above 1 year of age. A newer insulin pump called MiniMed 780G (Medtronic, Northridge, CA, USA) is approved for use in children above 7 years of age and has an advanced algorithm with automatic correction of boluses, fewer alarms, and simple operation compared to MiniMed 670 [88]. Several studies have demonstrated that the hybrid closed-loop system can reduce the frequencies of hypoglycemia and severe hypoglycemia in adults, children, and adolescents with type 1 diabetes, especially during nighttime concomitant with improving glycemic control [89,90,91,92,93,94]. Bergenstal et al. [90] reported a 0.5% reduction of HbA1c levels in patients with higher HbA1c levels at baseline benefitting most and a 44% reduction in TBR (glucose level less than 70 mg/dL (3.9 mmol/L)) with a 40% decline in dangerous hypoglycemia of glucose level less than 50 mg/dL (28 mmol/L). A 16-week, multicenter, randomized, open-label trial of children aged 6–13 years old who had type 1 diabetes demonstrated that the glucose level was in the target range (70–180 mg/dL) (3.9–10.0 mmol/L) for a greater percentage of time with the use of the closed-loop system than with the use of SAP, i.e., the target range increased from 53 ± 17% at baseline to 67 ± 10% in the closed-loop group and from 51 ± 16% to 55 ± 13% in the SAP group (*p* < 0.001). In both groups, TBR was similarly low (1.6% in the closed-loop group and 1.8% in the SAP group) [93]. In a multinational, randomized, crossover trial (Fuzzy Logic Automated Insulin Regulation [FLAIR]), patients aged 14–29 years old showed improved glycemic control without increasing hypoglycemia, i.e., mean difference of time with glucose levels below 54 mg/dL (<3.0 mmol/L) (advanced hybrid closed-loop system minus 670G system) −0.06% (95% CI −0.11 to −0.02); *p* < 0.0001 for non-inferiority) [94]. On the other hand, improved glycemic variability, especially during nighttime, with reduced hypoglycemia potentially improves sleep and the quality of life in children and their parents [95]. Individuals with impaired awareness of hypoglycemia also potentially improve their hypoglycemia awareness with the hybrid closed-loop system [96]. These results suggest that the use of the hybrid closed-loop system is generally effective and safe, especially in the nighttime, and decreases the burden of glycemic management overnight. The hybrid closed-loop system must make a significant impact on improving glycemic control with a reduction of hypoglycemia as well as minimizing the occurrence of severe hypoglycemia in children and adolescents with type 1 diabetes [88]. 

## 5. Conclusions

The reduction of hypoglycemia, especially the minimization of severe hypoglycemia, is a goal in the management of children and adolescents with type 1 diabetes. Evaluating the risk factors for developing severe hypoglycemia is a matter of great importance for preventing the occurrence of dangerous hypoglycemia. The new concept of TIR is currently used to evaluate glucose variability, glucose trends, and the quality of glycemic control [68]. Achieving the target range of TIR (glucose level of 70–180 mg/dL (3.9–10.0 mmol/L)) in more than 70% with a reduction of TBR (glucose level of less than 70 mg/dL (3.9 mmol/L)) less than 4% and minimizing dangerous hypoglycemia (glucose level of less than 54 mg/dL (3.0 mmol/L)) less than 1% is crucial in the management of type 1 diabetes [68]. This can be achieved through advanced diabetes technologies including CGM and the hybrid closed-loop system, even in pediatric patients.

## Figures and Tables

**Figure 1 jcm-12-00781-f001:**
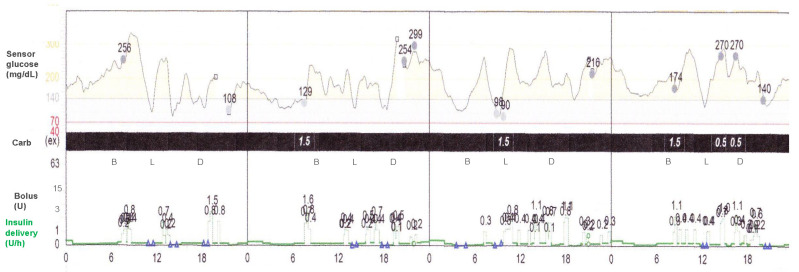
Sensor-augmented insulin pump with the predictive low-glucose suspend system (MiniMed 640G System, Medtronic, Northridge, California). Blue triangle marks indicate the predictive low-glucose suspension, i.e., suspension of basal insulin infusion with the hypoglycemia prediction algorithm. B: breakfast, L: Lunch, D: Dinner.

**Figure 2 jcm-12-00781-f002:**
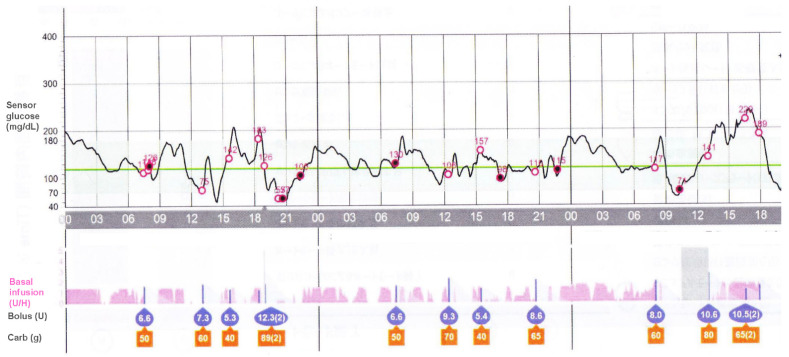
Closed-loop insulin delivery system (MiniMed 670G System, Medtronic, Northridge, California). The auto mode function increases, decreases, or stops the basal insulin infusion automatically (pink bars) in response to the sensor glucose readings to achieve the target glucose level of 120 mg/dL. The pump users should administer bolus insulin doses according to the carbohydrate counting at each meal.

**Table 1 jcm-12-00781-t001:** Incidence of severe hypoglycemia in children and adolescents with type 1 diabetes over time.

Report	Year	Incidence *	Reference No.
DCCT	1984–1993		[15]
Conventional		18.7	
Intensive		61.2	
Bulsara MK	1992	7.8	[16]
	2002	16.6	
Rewers A	1996–2000	19.0	[17]
O’Connell SM	2001	17.3	[18]
	2006	5.8	
Karges B	1995 2012	20.7 3.6	[19]
Urakami T	2003–2013	4.0	[20]
Cherubini V	2011–2012	7.7	[21]

* 100 persons/year.

**Table 3 jcm-12-00781-t003:** Risk factors for developing severe hypoglycemia.

Younger aga
Nocturnal hypoglycemia
Impaired awareness of hypoglycemia
Frequent episodes of hypoglycemia
Glycemic control (Recently, association between glycemic control and the risk of severe hypoglycemia seems to be weakened.)

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
