# Peer review of "The Advanced Diabetes Technologies for Reduction of the Frequency of Hypoglycemia and Minimizing the Occurrence of Severe Hypoglycemia in Children and Adolescents with Type 1 Diabetes"

_jcm, 2023, doi:10.3390/jcm12030781_

Round 1

Reviewer 1 Report

In the manuscript entitled “How can we reduce hypoglycemia with minimizing severe hypoglycemia in children and adolescents with type 1 diabetes?” Urakami reviewed advanced diabetes technologies for decreasing frequency of hypoglycemia and severe hypoglycemia in children and adolescents with T1D, such as, continuous glucose monitoring, the sensor-augmented insulin pump with the low-glucose suspend system and the hybrid closed-loop system. Although this manuscript is well written and properly organized, it is not sufficiently comprehensive for a review article. Therefore, I suggest to expand this MS by performing a very detailed, and comprehensive literature surveying, and to cite newly published papers.  

Minor points:

Spelling error, page 5, line 148 “sems”

Page 5, line 134, references “21,22,23” should be cited “21-23”

Author Response

January 9, 2023

Dear Dr. Sunny Huang
Assistant Editor, MDPI

Journal of Clinical Medicine

Thank you very much for reviewing the Manuscript ID: jcm-2122814 entitled “How can we reduce hypoglycemia with minimizing severe hypoglycemia in children and adolescents with type 1 diabetes?” and giving us a valuable comment. I answered the reviewers’ comments and revised the manuscript according to them.

I hope you would kindly review the revised manuscript and consider it for publication in Journal of Clinical Medicine.

Sincerely,

Tatsuhiko Urakami

Department of Pediatrics and Child Health, Nihon University School of Medicine, 30-1, Oyaguchi, Kami-Cho, Itabashiku, Tokyo, 173-8610, Japan

Reviewer 1

  • Although this manuscript is well written and properly organized, it is not sufficiently comprehensive for a review article. Therefore, I suggest to expand this MS by performing a very detailed, and comprehensive literature surveying, and to cite newly published papers. 

Reply

Thank you very much for your comment. According to your comment, I expanded the manuscript by a detailed and comprehensive literature survey and by citing newly published papers (I significantly rewrote this review article with citing new published papers) .

  • Spelling error, page 5, line 148 “sems”

Reply: I changed to “seems”

  • Page 5, line 134, references “21,22,23” should be cited “21-23”

Reply: I changed to “21-23”.

Reviewer 2 Report

The manuscript is a review exploring the possibilities for reduction of hypoglycemia, especially severe hypoglycemia, in children and adolescents with type 1 diabetes. The comments regarding the manuscript are written below.

-The title should be rewritten in order to become more informative. Mainly, the advanced diabetes technologies were studied for the reduction of the hypoglycemia frequency. This information should be included in the title. The word “we” could be omitted. 

-Table 1 needs to be expanded with additional data. In the title “children and adolescents” should be included. Several contents should be explained in detail (conventional-meaning?, intensive-meaning?); possibly in the legend of the table. 

-I suggest including a table or figure summarizing the main consequences of severe hypoglycemia. Also risk factors for developing severe hypoglycemia could be included. 

-I suggest using both values for different units of glucose levels (mg/dL and also mmol/L). 

-Hybrid closed-loop system: the prevention of hypoglycemia could also be made by selecting different glucose targets. This information should be included in the manuscript.

Author Response

January 9, 2023

Dear Dr. Sunny Huang
Assistant Editor, MDPI

Journal of Clinical Medicine

Thank you very much for reviewing the Manuscript ID: jcm-2122814 entitled “How can we reduce hypoglycemia with minimizing severe hypoglycemia in children and adolescents with type 1 diabetes?” and giving us a valuable comment. I answered the reviewers’ comments and revised the manuscript according to them.

I hope you would kindly review the revised manuscript and consider it for publication in Journal of Clinical Medicine.

Sincerely,

Tatsuhiko Urakami

Department of Pediatrics and Child Health, Nihon University School of Medicine, 30-1, Oyaguchi, Kami-Cho, Itabashiku, Tokyo, 173-8610, Japan

Reviewer 2

  • The title should be rewritten in order to become more informative. Mainly, the advanced diabetes technologies were studied for the reduction of the hypoglycemia frequency. This information should be included in the title. The word “we” could be omitted. 

Reply

I changed the title to “The advanced diabetes technologies for reduction of the frequency of hypoglycemia and minimizing the occurrence of severe hypoglycemia in children and adolescents with type 1 diabetes”

  • Table 1 needs to be expanded with additional data. In the title “children and adolescents” should be included. Several contents should be explained in detail (conventional-meaning?, intensive-meaning?); possibly in the legend of the table. 

Reply

I changed the title of Table 1 to “Incidence of severe hypoglycemia in children and adolescents with type 1 diabetes over time. I added the explanations in the legend of the table: i.e., Conventional; conventional insulin treatment: insulin injections twice a day with a mixture of regular and intermediate acting insulin, Intensive; intensive insulin treatment: multiple injections of insulin with basal-bolus regimen.

  • I suggest including a table or figure summarizing the main consequences of severe hypoglycemia. Also risk factors for developing severe hypoglycemia could be included. 

Reply: Main consequences of severe hypoglycemia is shown in new Table 2, and risk factors for developing severe hypoglycemia is shown in new Table 3.

4) I suggest using both values for different units of glucose levels (mg/dL and also mmol/L). 

Reply: I added the unit of mmol/L for glucose levels and HbA1c values.

  • Hybrid closed-loop system: the prevention of hypoglycemia could also be made by selecting different glucose targets. This information should be included in the manuscript.

Reply

I added the information of the glucose targets such as “below 70 mg/dL (3.9 mmol/L), below 50 mg/dL (28 mmol/L) etc.” for different studies in the sections of CGM, SAP and the hybrid closed-loop system.

Round 2

Reviewer 1 Report

Author answered to all reviewers comments, therefore I suggest to accept MS in present form. 

Reviewer 2 Report

The revised manuscript is improved. I have no further comments.